# Spatial variations and determinants of timely completion of vaccination in Ethiopia using further analysis of EDHS 2019 data: Spatial and multilevel analysis

Muluken Chanie Agimas[1]*, Aysheshim Kassahun Belew[2], Mekonnen Sisay[2], Lemlem Daniel Baffa[2], Moges Gashaw[3], Zufan Yiheyis Abriham[4], Esmael Ali Muhammad[2], Zeamanuel Anteneh Yigzaw[5], Berhanu Mengistu[2]

1 Department of Epidemiology and Biostatistics, Institute of Public Health, College of Medicine and Health Science, University of Gondar, Gondar, Ethiopia, 2 Department of Human Nutrition, Institute of Public Health, College of Medicine and Health Sciences, University of Gondar, Gondar, Ethiopia, 3 Department of Physiotherapy, School of Medicine, College of Medicine and Health Sciences, University of Gondar, Gondar, Ethiopia, 4 Department of Medical Parasitology, School of Biomedical and Laboratory Sciences, College of Medicine and Health Sciences, University of Gondar, Gondar, Ethiopia, 5 Department of Health Promotion and Behavioral Science, School of Public Health, College of Medicine and Health Sciences, Bahir Dar University, Bahir Dar, Ethiopia

* mulukensrc12@gmail.com

**Data Availability Statement:** All relevant data is within the manuscript.

## Abstract

### Background

Timely vaccination is the practice of administering the vaccine within the first birthday of the child. Not vaccinating the child at the appropriate age is the cause of improper protection of diseases and can be a possible factor in death. The problem of not completing the vaccine in the scheduled period is a globally distributed problem, but especially in sub-Saharan African countries, it is a bottleneck to child health. Even if timely vaccination is crucial for reducing the impact of VPDs, there are no current national-level studies to generate conclusive and tangible evidence in Ethiopia.

### Objective

To assess spatial variations and determinants of timely completion of vaccination in Ethiopia using further analysis of EDHS 2019 data.

### Method

The secondary data analysis of a community-based cross-sectional study design was employed among 3094 participants. Stata-14 software was used for data cleaning, recording, and analysis. Arc GIS version 10.3 and Kuldorff SAT scan version 9.6 software are used for spatial and SAT scan statistics. A multilevel mixed-effect binary logistic regression analysis was used to identify the predictors of timely vaccination. The clustering effect was also evaluated by Moran's I statistics and intra class correlation.

**Funding:** The author(s) received no specific funding for this work.

**Competing interests:** The authors have declared that no competing interests exist.

**Abbreviations:** ANC, Antenatal Care; AOR, Adjusted Odds Ratio; BCG, Bacilli Calamite Guerin; CI, Confidence Interval; DHS, Demographic Health Survey; EDHS, Ethiopian Demographic Health Survey; OR, Odds Ratio; OPV, Oral polio Vaccine; PCV, . . .. Pneumococcal conjugated Vaccine; VPDs, Vaccine Preventable Diseases; WHO, World Health Organization.

## Results

The timely completion of vaccination among Ethiopian women who had a child aged 12–35 months was 19.5% (95%CI: 18.2–20.8), and the spatial distribution of timely completion of vaccinations in Ethiopia was non-randomly distributed. A statistically significant high proportion of timely completion areas were clustered in the eastern part of Amhara, the south part of Afar, Addis Ababa, and Oromia. The primary cluster was located at a 13.11 km radius in Diredawa, which was 3.68 times higher than outside the window (RR = 3.68, LLR = 68.76, p-value < 0.001). History of antenatal care follow-up (AOR = 1.63, 95% CI: 1.3–2.04), giving birth at health facilities (AOR = 1.63, 95% CI: 1.25–2.13), age ≥ 35 years (AOR = 186, 95% CI: 1.35–2.63), age 25–34 years (AOR = 1.72, 95% CI: 1.33–2.21), and being richest (AOR = 2.71, 95% CI: 1.86–3.94) were the factors contributing to the timely completion of vaccination.

## Conclusion

The prevalence of timely completion of vaccination was low in Ethiopia, and the spatial distribution of timely completion of vaccination in Ethiopia was non-randomly distributed across the regions. The factors associated with the timely completion of vaccinations were ANC follow-up, place of delivery, age of the participant, and wealth index. We recommend expanding facility delivery, antenatal care services, and empowering women to scale up timely vaccination in Ethiopia.

## Introduction

According to the World Health Organization, timely vaccination is the practice of administering the vaccine within the first one-year schedule [1]. When the vaccines are taken within the time frame or at the scheduled time, their effectiveness increases [2]. Even though each year 2 to 3 million deaths are reversed by the timely vaccine, still 1.5 million children die due to vaccine-preventable diseases [3]. Not vaccinating the child at the appropriate age is the cause of improper protection of diseases and can be a possible factor in death [4, 5]. Based on this fact, the World Health Organization (WHO) suggests that timely vaccination provides the full benefit of the vaccine's potency [6]. Not timely vaccination also includes giving a vaccine before vaccination, and this can also fail to protect the disease properly because of the suboptimal seroconversion rate [7].

Timely vaccination is one of the pillars of vaccination program, but most of the time it is an ignored indicator of vaccination program performance [8]. The problem of not completing the vaccine in the scheduled period is a globally distributed problem, but especially in sub-Saharan African countries, including Ethiopia, the problem is a bottleneck in child health [9].

In low- and middle-income countries, so many challenges contribute to untimely vaccination practices. Among these challenges, the problem of supply chain management, health services, and the performance of the health care provider is the most common factor [10]. Even though the factors of timely vaccination are very contextual [11], home delivery [12], low-education attainment and below four antenatal care visits [13], unplanned pregnancy and child male sex [3], highest mothers/caregivers age [14], vaccine hesitancy [15], being a multiparous mother [16], and rural children and poorest quintile are also affected the timely vaccination [10].

To tackle the problem of untimely vaccinations, Ethiopia has key strategic activities. But still, the mortality of children caused by vaccine-preventable diseases (VPD) is high, and, according to the WHO-2019 report, Ethiopia is the 5th leading country in untimely vaccination [17]. In Ethiopia, many studies are conducted on the coverage of full vaccination practice, but studies on timely vaccination and associated factors are few. Among such, a few studies report timely vaccination prevalence; Addis Ababa 55.9% [15], in 2015, pastoralist Woredas of Ethiopia 78.1% [14]. Even if timely vaccinations are very crucial for reducing the impact of VPDs, there are no current national-level studies to generate conclusive and tangible evidence about the determinants of timely vaccination in Ethiopia for policymakers, planners, non-governmental organizations, health facilities, and educators. Thus, the aim of this national-level study is to determine the pooled prevalence and identify associated factors of timely completion of vaccination practice in Ethiopia using the 2019 Ethiopian demographic health survey.

## Methods

### Study area, study design, and period

A community-based cross-sectional study design was used among Ethiopian childbearing women from March 21, 2019 to June 28, 2019 [18]. Ethiopia is a sub-Saharan African country, and based on February 26, 2023, world meter reports, Ethiopia has a total population of 123,001,400, and around 21.3% of the population lives in urban areas [19]. Ethiopia comprises the following regions: Afar, Somalia, South Africa, and the nationality of people in the region: Amhara, Harari, Tigray, Benishangul Gumuz, Gambela, and Oromia. Additionally, there are two self-administrative cities, namely Addis Ababa and Diredawa.

### Source population and study population

The source population was all women in Ethiopia who had a child aged 12–35 months earlier in the national survey, and women who had a child aged 12–35 months in the selected enumeration were the study population. All women who had a child aged 12–35 months during the survey were included in the study [18].

### Variables

**Dependent variable.**  Timely completion of vaccination (Yes, No).

### Independent variables

**Socio-demographic characteristics of the women.**  residence, wealth index, women's age, educational level, region, sex of the child, sex of household head, having a cell phone, listening to the radio.

**Obstetrics and other health service utilization-related factors.**  current breastfeeding, total numbers of live births, ANC utilization (yes or no), frequency of ANC utilization (<4 or ≥4 visits), and current pregnancy status.

### Operational definitions

**Timeliness of vaccination.**  A child is considered "yes" for timely vaccination if the child received BCG within the first 4 weeks, received OPV1, Penta 1, PCV1, and Rota 1 from 6 weeks to 10 weeks, received OPV 2, Penta 2, PCV 2, and Rota 2 from 14 weeks to 18 weeks, and received measles vaccination from 9 to 10 months. On the contrary, a child is classified as "no" for timely vaccination if they received at least one dose of the vaccine below the above

minimum recommended age for each antigen and/or when they received at least one dose of the vaccine above the maximum recommended age [1, 20, 21].

## Sampling procedure and sampling technique

In the 2019 EDHS survey, a total of 9160 households were employed for the sample, and of these, 8794 were occupied, whereas 8663 participated in the survey. This sample was stratified based on urban and rural residences. Samples of enumeration areas were nominated from each stratum, and a total of 21 sampling strata were formed. Based on a well-prepared sampling frame for each stratum of lower administrative levels, the proportional allocation technique was used. Within 25 enumeration areas, equal allocation of the samples was done to improve precision across the region of Ethiopia. All women aged 15–49 who were either permanent residents of the selected households or visitors who slept in the household the night before the survey were eligible to be interviewed, and 305 clusters were incorporated into the study [18]. Finally, 3094 participants who had children aged 12–35 months were interviewed about the timely completion of vaccinations. A two-stage cluster sampling was employed to select the participants.

## Data collection procedures and data quality assurance

Initially, the data were collected in the 2019 mini-EDHS, and for the purpose of the current research, data were requested online and retrieved from the international program of demographic at https://www.dhsprogram.com/data/dataset_admin/login_main.cfm) by explaining the main purpose of the data request. After two consecutive working days, the data set was released. The data collection method of the 2019 mini-EDHS was an interview-administered method using a structured and pre-tested questionnaire from Ethiopian regions among women of childbearing age 12–35 months. After the data were accessed, extraction of the dependent and independent variables was done. To assure the quality of the data, training was conducted for data collectors, supervisors, field editors, and reserve interviewers. The pretest was also conducted to validate and assure the reliability of the tool [18].

## Data processing and analysis

After the data were extracted from the EDHS-2019 mini data set, Stata-14 software was used for data cleaning, recording, computing, transforming, and a multilevel mixed-effect binary logistic regression by:

Logit $(Y_{ij})$ = $\beta_{0j}$ + $\Sigma\beta X_i$ + $\Upsilon Z_j$ + $\varepsilon_j$, where $\beta_{0j}$ = $\beta_0$ + $\mu_j$, $\mu_j \sim N(0, \sigma^2 u)$ and $\varepsilon_j$ = $\varepsilon_0$ + $\varepsilon_j$, $\varepsilon_j \sim N(0, \sigma^2 \varepsilon)$ [22].

Where $(Y_{ij})$ = a probability of timely vaccination practice,

"i" = enumeration area, rural or urban region, "j."

"$\beta_{0j}$" = cluster random intercept,

"$\varepsilon_j$" = residual for each cluster "j",

"$\beta$" = the fixed effect regression coefficient,

"$X_i$" = level-1 predictors,

"$\Upsilon Z_j$" = level-2 factors in cluster j.

In mixed effect binary logistic regression, logit $(Y_{ij})$ = ln $(Y_{ij}/(1-Y_{ij}))$ is the so called log-odds (logit link) for timely vaccination practice. We used the mixed-effect binary logistic regression model because of the hierarchal nature of the EDHS data. That means individuals (women) are nested within households, and the households are nested within the clusters. It is impossible to manage the hierarchical data or show the clustering effect using classical (traditional) binary logistic regression. For all analysis procedures, sampling weight was used for

**Table 1. Multilevel mixed effect binary logistic regression model of individual and community level factors predicting timely completion of vaccination in Ethiopia using EDHS 2019 data.**

| Random effect | Null model | Model I | Model II | Model III |
|---|---|---|---|---|
| Variance | 0.912 | 0.488 | 0.656 | 0.2798 |
| ICC | 21.7% | 12.92% | 16.6% | 7.8% |
| PCV (%) | Reference | 46.5% | 28.1% | 69.3% |
| Log likelihood | -989.56 | -812.2 | -865.56 | -778.2 |
| AIC | 15807 | 14428 | 15855 | 13262 |

complex surveys and an unequal probability of selection. A multilevel mixed-effects logistic regression analysis was used to identify the possible factors associated with the timely completion of vaccinations because of the hierarchical nature of the data. The "Svy" command was used as t the sampling weight of cluster sampling. Intraclass coefficient (ICC = σ2ε/σ2ε+σ2μ; Where σ2μ = π/23, σ2ε = between-group variance) was used for the measurement clustering effect [23] which was 21.7%. For all models (null model, model I, and model II), ICC, a proportional change in variance (PCV = Variance of the null model-Variance of model-i/Variance of null model*100) [24], log-likelihood test, and Akaike information criterion (AIC = 2k-2lnĹ, where k = the numbers of parameter, Ĺ = the maximum value of the likelihood function of the model) model was calculated. The best model was selected by AIC or using the lowest value of AIC (**Table 1**). Variables with a p-value of less than 0.05 and a 95% confidence level were considered for statistical significance.

## Spatial autocorrelation

To analyze Moran's I, ArcGIS version 10.3 was used. The global Moran's I work on the value of 1 to -1. A value close to −1 shows dispersed age-appropriate vaccination practice. Whereas Moran's I value closest to +1 shows clustered timely vaccination practice, and the value 0 shows randomly distributed timely vaccination practice. Moran's I (p-value < 0.05) was used to declare the statistical significance of spatial auto-correlation.

## Hot spot analysis and spatial interpolation

The spatial autocorrelation between the regions of Ethiopia was assessed by hot spot analysis (Getis-Ord Gi* statistic). The significance of clustering was evaluated by a Z-score with a p-value of 0.05 [25]. Because of a lack of resources and time, it is impossible to find timely vaccinations in all parts of the country. For this reason, predicting the unsampled area based on sampled data is a very crucial issue. To achieve this aim, the ordinary kriging spatial interpolation technique was employed. Principally, closer things are more related than distant things. Therefore, the spatial dependence on the timely completion of vaccinations was considered. The false discovery rate correction was used during the hot spot analysis.

## Spatial scan statistics

Bernoulli-based model spatial scan statistics using Kuldorff Sat Scan version 9.6 software were used to evaluate the statistical significance of the timely completion of vaccination in Ethiopia [26]. The default maximum spatial cluster size of 50% was used.

## Ethical approval and consent to participate

Because it was secondary data, ethical consent was not applicable; rather, data was requested and authorized to be accessed by the DHS International at:

https://www.dhsprogram.com/data/dataset_admin/login_main.cfm. The EDHS 2019 data
was collected in accordance with national and international ethical standards. The authoriza-
tion letter was obtained to use the data.

## Results

### Socio-demographic characteristics

A total of 3094 women who had 12–35 months of the child preceding the survey were included
and interviewed about the timely completion of vaccinations. About 1636 (52.9%) were within
the age group of 25–34 years, and 1532 (49.5%) and 1161 (37.5%) of women had no education
or primary education, respectively. Furthermore, the majority of the participants, 2299
(74.3%), were from rural residences (**Table 2**).

### Obstetrics and other health service utilization-related factors

In this study, the majority of women, 2562 (62.7%), gave birth at home, and 3000 (73.5%) and
553 (13.5%) of them were not pregnant or didn't know about their pregnancy status, respec-
tively, during the survey. About 2629 (85%) had a history of ANC follow-up, and 1669 (54%)

**Table 2.** Socio-demographic characteristics of the participants who had children aged 12–35 months in Ethiopia based on 2019 EDHS data.

| Variables | Category | Weighted Frequency | Percent (%) |
|---|---|---|---|
| Sex of child | Male | 1562 | 50.5% |
| | Female | 1532 | 49.5% |
| Women age | 15–24 | 857 | 27.7% |
| | 25–34 | 1636 | 52.9% |
| | > = 35 | 601 | 19.4% |
| Sex of head of the household | Male | 2695 | 87.1% |
| | Female | 399 | 12.9% |
| Education status | No education | 1532 | 49.5% |
| | Primary school | 1161 | 37.5% |
| | Secondary | 268 | 8.7% |
| | Higher education | 132 | 4.3% |
| | Diredawa | 17 | 0.5% |
| Region | Tigray | 213 | 6.9% |
| | Afar | 49 | 1.6% |
| | Amhara | 614 | 19.8% |
| | Oromia | 1236 | 40% |
| | Somalia | 201 | 6.5% |
| | Benishangul | 37 | 1.2% |
| | SNNPR | 609 | 19.7% |
| | Gambela | 14 | 0.4% |
| Residence | Urban | 795 | 25.7% |
| | Rural | 2299 | 74.3% |
| Use mobile | Yes | 995 | 24.4% |
| | No | 3088 | 75.6% |
| | Poorest | 693 | 22.4% |
| Wealth index | Poorer | 662 | 21.4% |
| | Middle | 598 | 19.3% |
| | Richer | 513 | 16.6% |
| | Richest | 628 | 20.3% |

**Table 3. Obstetrics and other health service utilization-related factors of timely completion vaccinations in Ethiopia using 2019 EDHS data.**

| Variables | category | Weighted Frequency | Percent (%) |
|---|---|---|---|
| Current breastfeeding | Yes | 716 | 23.1% |
| | No | 2378 | 76.9% |
| Listen radio | Yes | 443 | 16% |
| | No | 2381 | 84% |
| Have cell phone | Yes | 1670 | 54% |
| | No | 1424 | 46% |
| | Don't know | 553 | 13.5% |
| Current pregnancy | Yes | 269 | 8.7% |
| | No/unknown | 2825 | 91.3% |
| Numbers of live birth | 1–2 | 1317 | 42.6% |
| | 3–4 | 859 | 27.8% |
| | >4 | 918 | 29.6% |
| | Yes | 2629 | 85% |
| ANC utilization | No | 465 | 15% |
| Frequency of ANC visit | <4 | 2925 | 94.5% |
| | > = 4 | 169 | 5.5% |
| Usual care taker of the child | mother | 2890 | 93.4% |
| | Other than mothers | 204 | 6.6% |
| place of delivery | Home | 1425 | 46% |
| | Health facility | 1669 | 54% |

of the participants gave birth at the health facilities. Furthermore, 1285 (31.4%) women had four children or more (**Table 3**).

## Overall age-appropriate vaccination practice

In the current study, the timely completion of vaccinations among Ethiopian women who had children aged 12–35 months was 19.5% (95%CI: 18.2–20.8) (**Fig 1**).

## Timely completion of vaccination for each vaccine dose

Based on EDHS-2019 data, the highest number of children (2148) took the polio-1 vaccine on time, followed by the Penta/DPT-1 vaccine (2023) (**Fig 2**).

## Spatial distribution of timely completion of vaccinations in Ethiopia

About 305 clusters were considered in the spatial analysis of the timely completion of vaccinations in Ethiopia. The high proportion of timely completion of vaccinations was represented by the red dots. The eastern part of Amhara, Tigray, and Addis Ababa had the highest timely completion of vaccination practice. Whereas the lowest proportion of timely completion of vaccination practice was reported in Afar, the northern part of Amhara, Oromia, and the Somali region as compared to the other parts of the Ethiopian regions (**Fig 3**).

## Spatial autocorrelation

As the evidence showed from the spatial autocorrelation analysis (global Moran's I test), the spatial distribution of timely completion of vaccinations in Ethiopia was non-randomly distributed. The spatial autocorrelation analysis of the global Moran's I test reported that Moran's

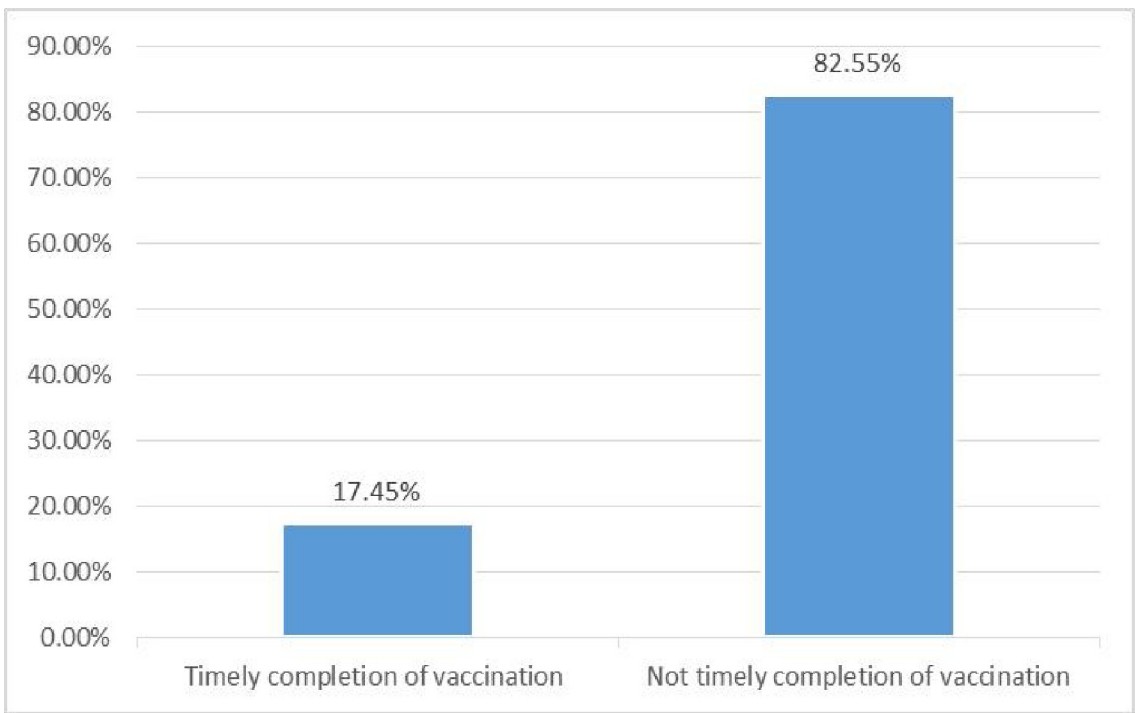

**Fig 1. Timely completion of vaccination among in Ethiopia based on EDHS-2019 data.**

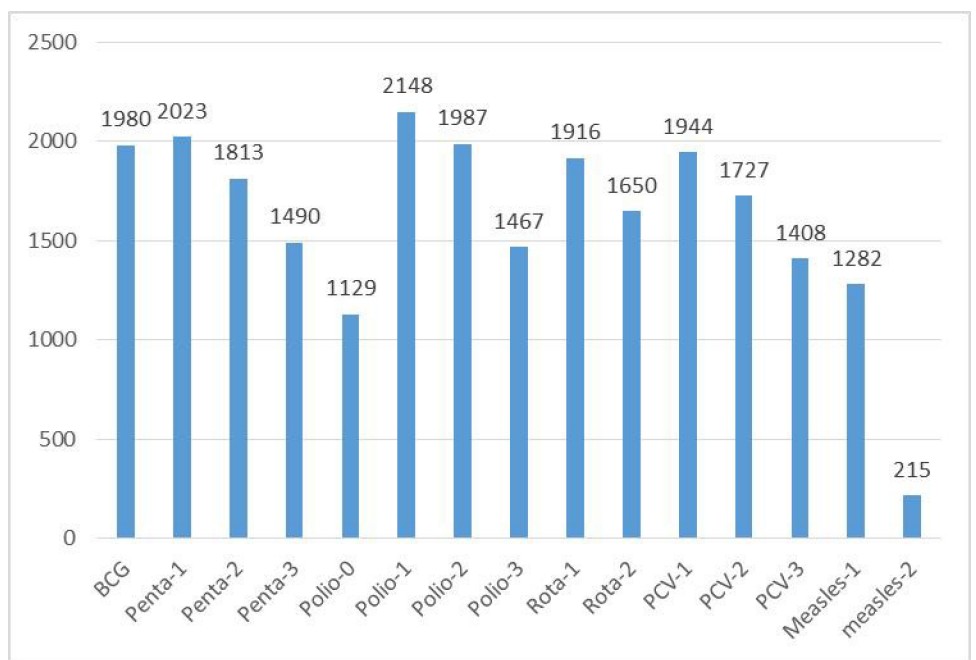

**Fig 2. Timely completion of vaccination for each vaccine dose in Ethiopia based on EDHS-2019 data.**

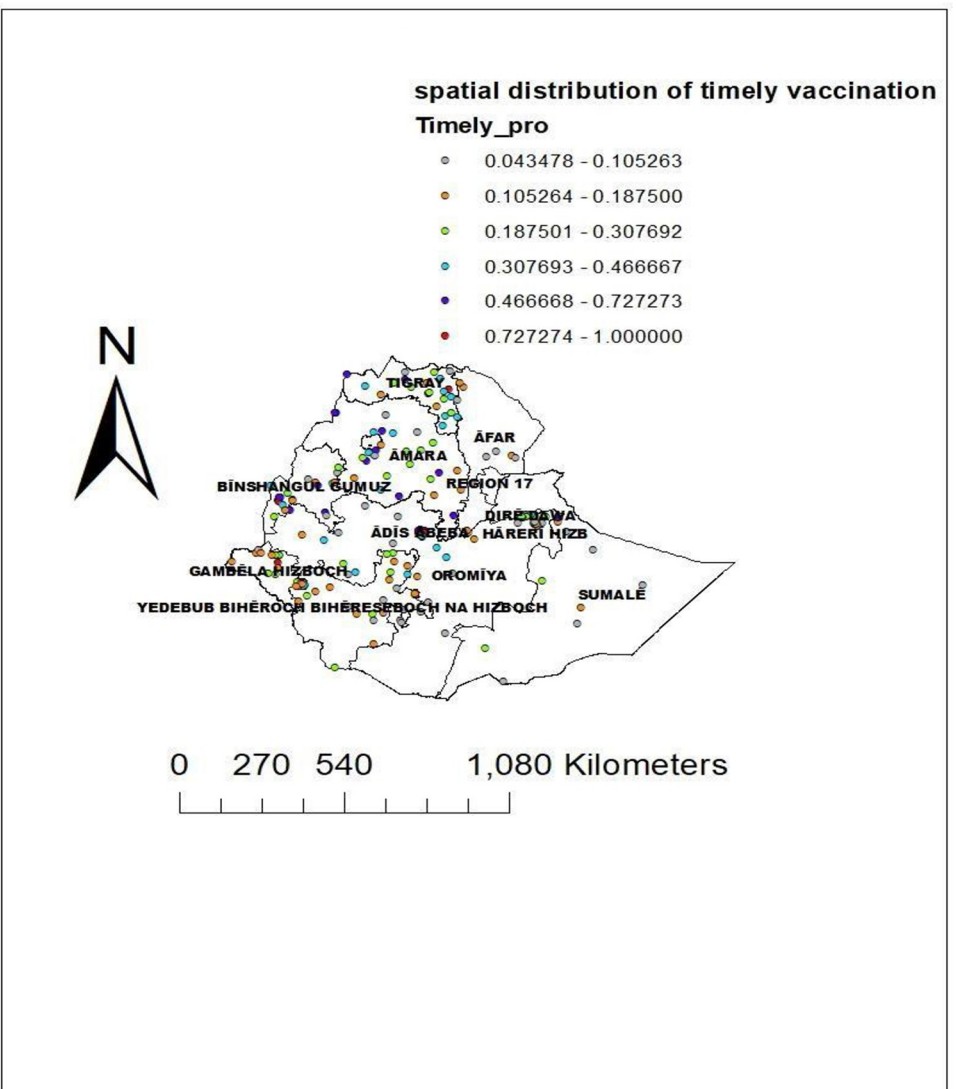

**Fig 3. Spatial distribution of timely completion of vaccination in Ethiopia using 2019 EDHS data.** Source: *https://africaopendata.org/dataset/ethiopia-shapefiles*.

index value was 1.114, the z-score value was 21.26, and its p-value was <0.001. This value showed that the timely completion of vaccinations in Ethiopia had statistically significant clustering across the regions. (**Fig 4**).

## Hot/cold spot area of timely completion of vaccinations in Ethiopia

According to the Getis-Ord Gi* spatial analysis, statistically hot spot areas or a high proportion of timely completion areas were clustered in the eastern part of Amhara, the southern part of Afar, Addis Ababa, and Oromia. Whereas the cold spot areas for timely completion of vaccinations in Ethiopia were the southern people and nationalities region (**Fig 5**). When conducting multiple comparisons, there is an increased probability of false positives. To manage this problem we used the false discovery rate correction during the hot spot analysis.

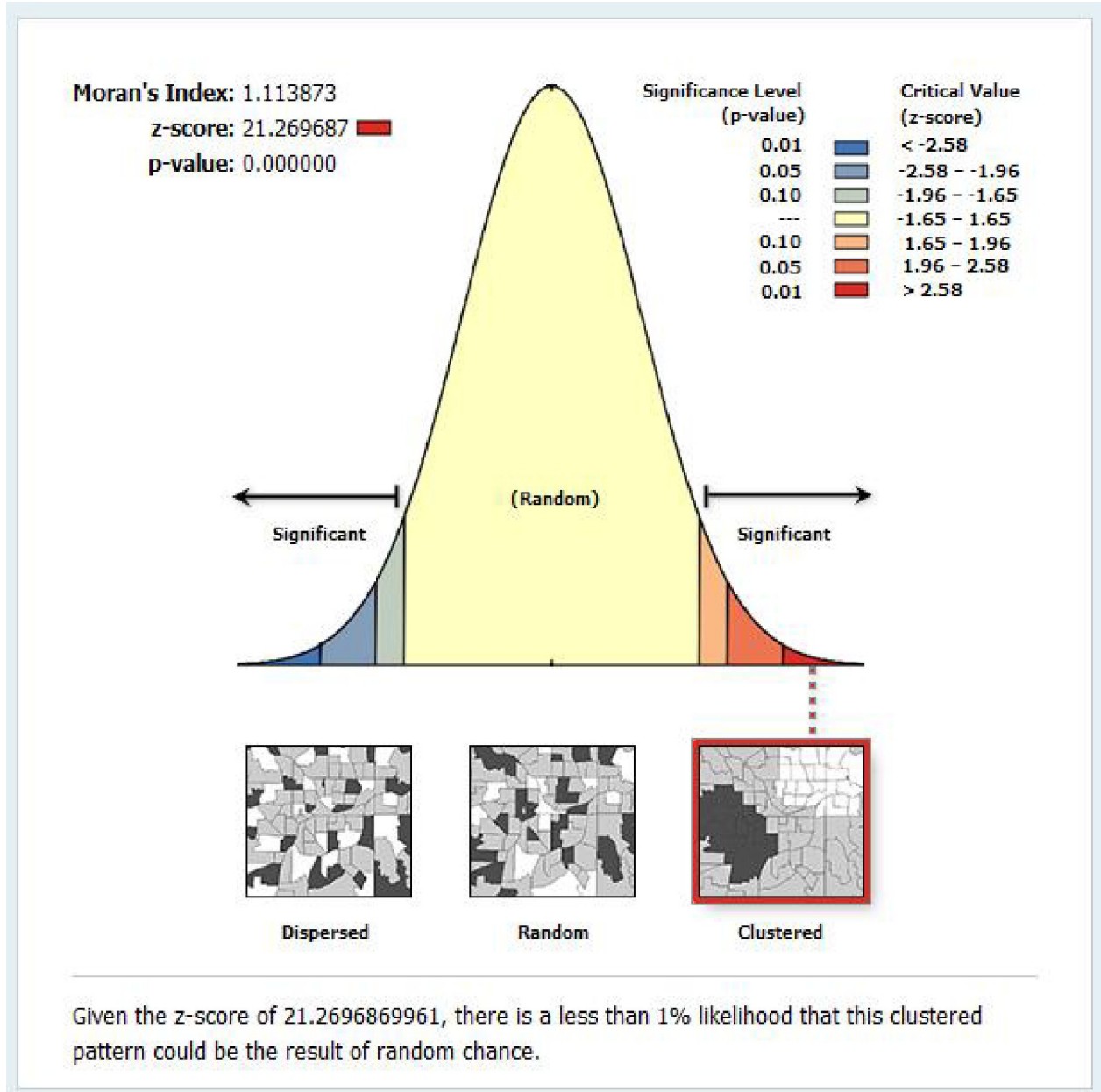

**Fig 4. Spatial auto-correlation of timely vaccination in Ethiopia using 2019 EDHS data.** Given the z-score of 21.2696869961, there is a less than 1% likelihood that this clustered pattern could be the result of random chance.

## Spatial interpolation

Using spatial ordinary kriging or spatial interpolation analysis, the highly predicted practices of timely completion of vaccinations in Ethiopia were the Afar region, eastern Amhara, Addis Ababa, Oromia, and Diredawa. On the other hand, the Somali south and eastern parts of the southern nation and the nationality of the people's region were predicted to have a low practice of timely completion of vaccinations (**Fig 6**).

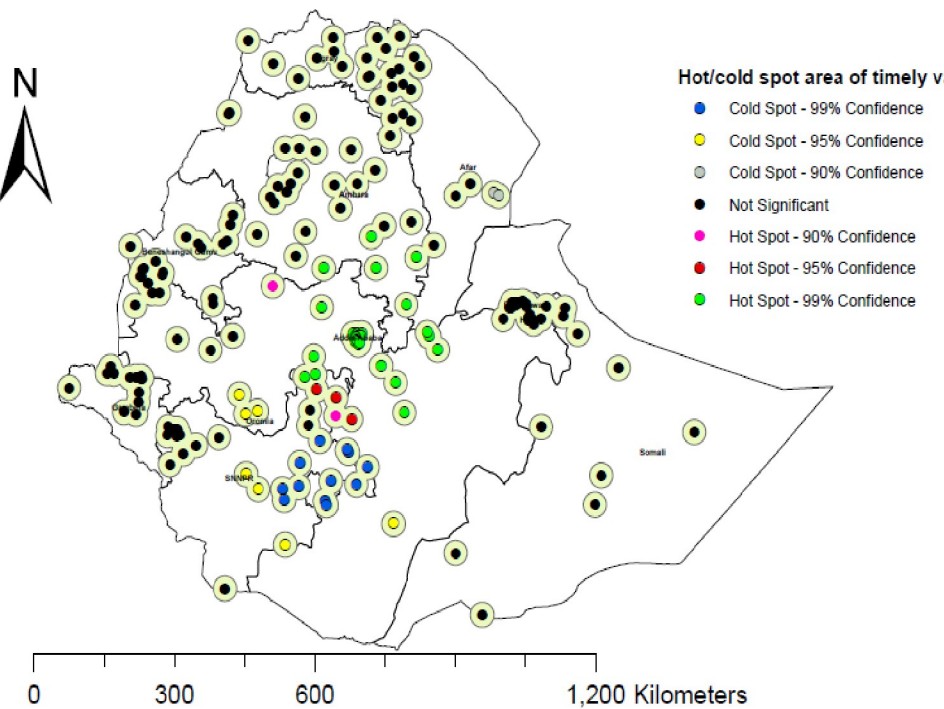

**Fig 5. Hot and cold spot area of timely completion of vaccination in Ethiopia using 2019 EDHS data.** Source: *https://africaopendata.org/dataset/ ethiopia-shapefiles*.

### Incremental autocorrelation

As the evidence showed in the incremental autocorrelation, the number of distance bands was 10. The bands started at 203 km, where the timely vaccination practice was detected or identified. This means that at a distance of 203 km with a statistically significant z-score value, the spatial clustering of timely vaccination was highly pronounced or prominent (**Fig 7**).

### Sat scan analysis

As **Fig 8** shows, the red window, the yellow window, and the red dots on the map were the significant clusters of timely completion of vaccinations in Ethiopia. The total number of clusters found to be statistically significant was 103. Of the total statistically significant clusters, 20 of them were primary (most likely) cluster types; the rest 83 clusters were secondary, tertiary, and quarterly clusters. The primary cluster was located at 9.066209 N and 38.754640 E within a 13.11 km radius in Diredawa. The timely completion of vaccination in the most likely clustered area was 3.68 times higher than outside the window (RR = 3.68, LLR = 68.76, P-value < 0.001) (**Table 4** and **Fig 8**).

### Factors of timely completion of vaccinations

Three models were run. These were the null model (model-0), model-I (model of the level predictors), model II (model with group-level variables), and model III (a combination of all levels

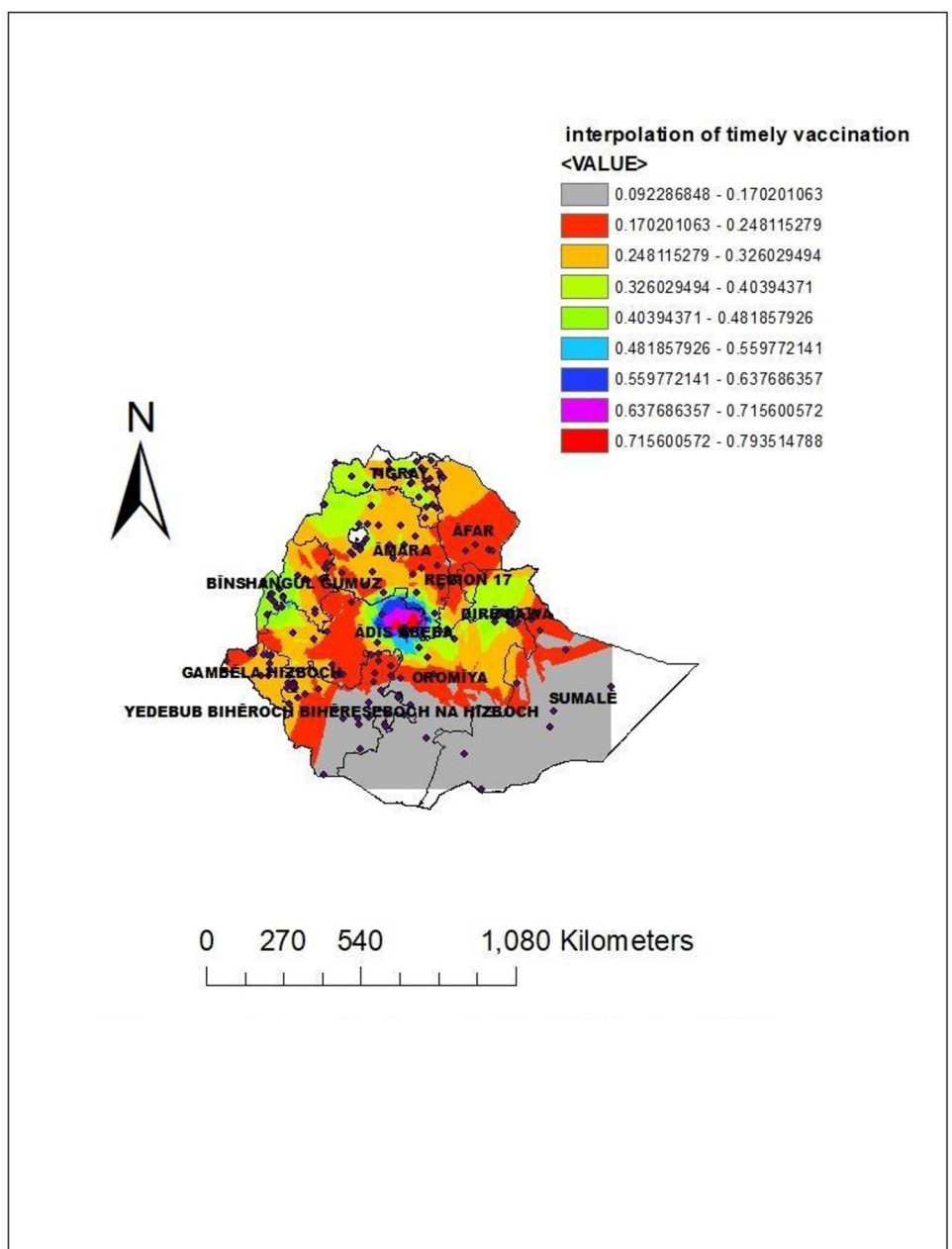

**Fig 6. Spatial interpolation of timely vaccination in Ethiopia using 2019 EDHS data.** Source: *https://africaopendata. org/dataset/ethiopia-shapefiles*.

of models). From these models, the best model was selected based on the criteria of lowest AIC and deviance. Based on these values, the lowest AIC and deviance and the best model were model III (**Table 1**). In the Bivariable multilevel binary logistic regression model, variables such as wealth index, ANC utilization history, place of delivery, current breastfeeding, educational status, and sex of household head were the potential candidate variables for multivariable mixed effect logistic regression at a p-value of less than 0.25. Finally, in multivariable mixed effect logistic regression analysis, the age of the women and wealth index were the significant factors of the level I predictor, and variables such as place of delivery were the

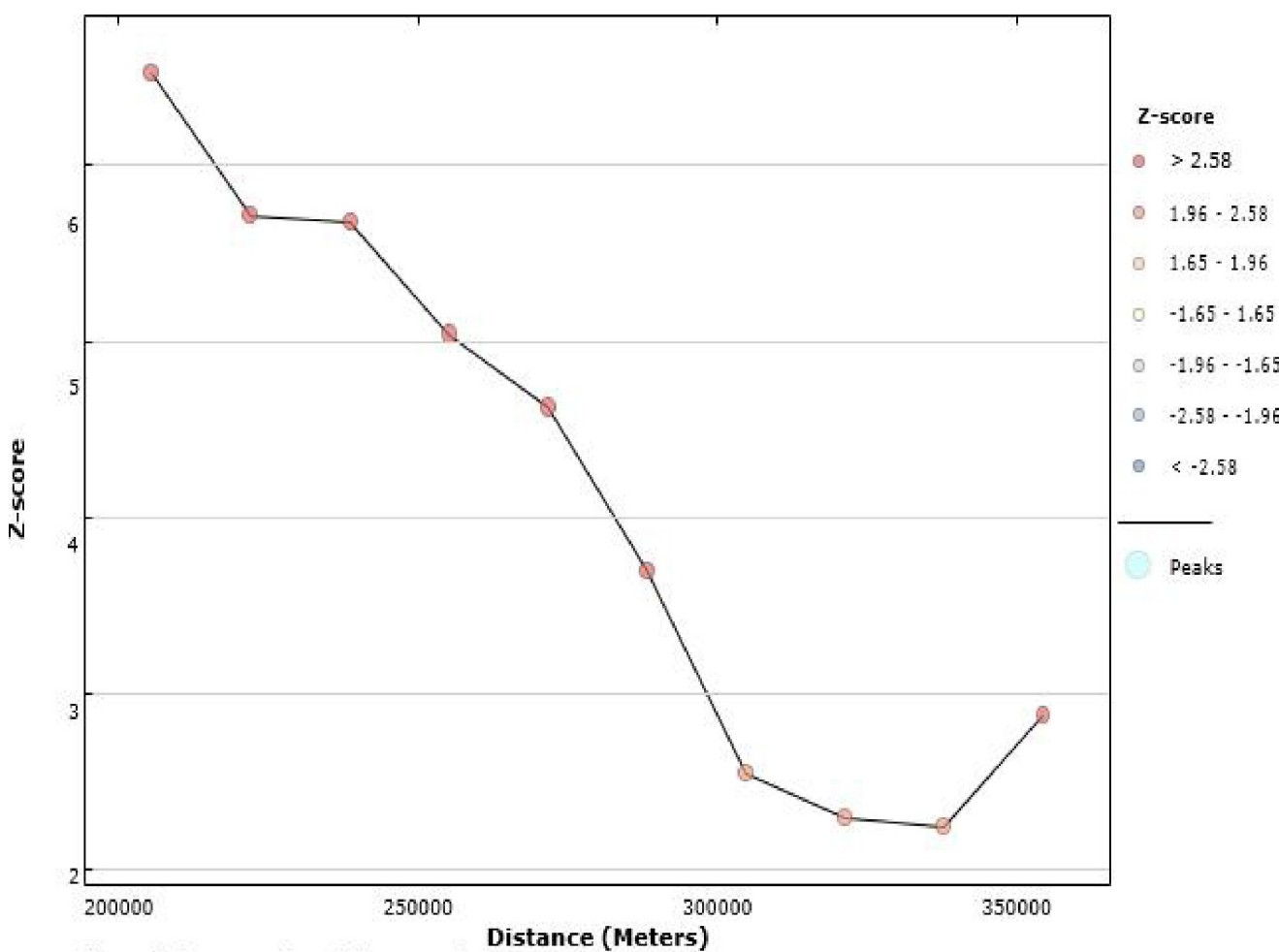

**Fig 7. Incremental spatial autocorrelation of timely completion of vaccination in Ethiopia, 2019, EDHS.**

significant factors of the level two predictor. As such, the odds of timely completion of vaccination among women who had a history of ANC follow-up were 1.63 times (AOR = 1.63, 95% CI: 1.3–2.04) higher than those with no history of ANC follow-up. Women who give birth at health facilities were 1.62 (AOR = 1.62, 95% CI: 1.25–2.13) times more likely to practice timely completion of vaccinations than those who give birth at home. Again, the odds of timely completion of vaccinations among women with ages ≥ 35 years and 25–34 years were 1.89 (AOR = 1.89, 95% CI: 1.35–2.63) and 1.72 (AOR = 1.72, 95% CI: 1.33–2.21) times higher than those aged 15–24 years old. Furthermore, the odds of timely completion of vaccinations among the richest women were 2.71 times (AOR = 2.71; 95% CI: 1.86–2.94) more likely than the poorest women (**Table 5**).

## Discussion

In this study, an attempt has been made to assess the spatial distribution of timely completion of vaccination and its determinants in Ethiopia using EDHS 2019 data. Thus, the prevalence of timely completion of vaccinations was 19.5% (95% CI: 18.2–20.8), and the significant factors for timely completion of vaccination were ANC follow-up, place of delivery, wealth index, and

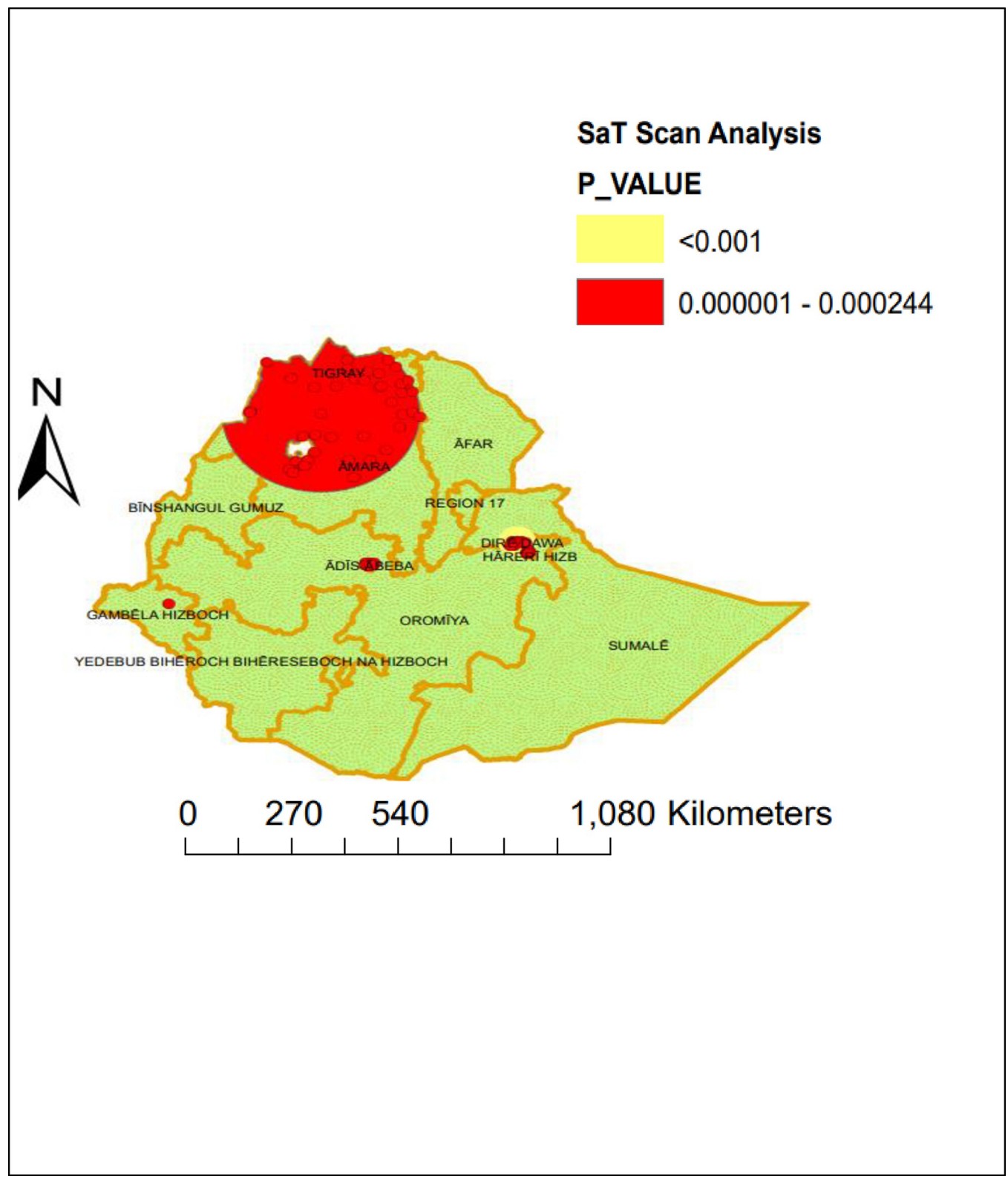

**Fig 8. Sat scan analysis of timely completion of vaccination in Ethiopia using EDHS 2019 data.** Source: *https://africaopendata.org/dataset/ethiopia-shapefiles*.

**Table 4. Significant clusters for timely completion of vaccination in Ethiopia using 2019 EDHS data.**

| Cluster | Significant Enumeration Areas (clusters) detected | Coordinate/radius | Population | Cases | RR | LLR | P-value |
|---|---|---|---|---|---|---|---|
| Primary | 256, 265, 266, 257, 258, 263, 267, 264, 262, 261, 260, 273, 276, 259, 268, 275, 271, 270, 269, 272 | (9.066209 N, 38.754640 E) / 13.11 km | 135 | 87 | 3.68 | 68.76 | <0.001 |
| Secondary | 289, 290, 297, 293, 291, 285, 292, 295, 294, 283, 288, 284, 286, 287, 282, 296, 302, 281, 303, 298, 301, 304, 305, 299, 300, 251, 253, 232, 231, 236, 238, 246, 237, 240, 233, 235, 243, 242, 239, 234 | (9.617215 N, 41.883048 E) / 44.04 km | 350 | 117 | 1.88 | 21.57 | <0.05 |
| Tertiary | 56, 82, 83, 84, 22, 9, 57, 78, 8, 21, 7, 59, 13, 1, 58, 74, 54, 2, 81, 14, 55, 12, 85, 23, 6, 61, 11, 75, 53, 62, 18, 20, 4, 5, 60, 17, 24, 16, 25, 10, 3, 19 | (12.939659 N, 37.743025 E) / 225.90 km | 418 | 125 | 1.67 | 15.16 | <0.05 |
| Quarterly | 212 | (7.986818 N, 34.544894 E) / 0 km | 14 | 12 | 1.67 | 14.41 | <0.05 |

age of the women. As well, the timely completion of vaccination was not randomly distributed across the regions of Ethiopia.

The prevalence of timely completion of vaccinations in Ethiopia is lower than in studies conducted in Gondar city (31.9%) [27], Toke Kutaye district of central Ethiopia 23.9% [20], in Afar, Ethiopia 43.7% [28], Uganda 45.6% [29], china 43.72%-59.25% [25], Tanzania 82.7% -88.5% [30], Saudi Arabia 73% [31], Ghana 87.3% [32], Cameroon 73.3% [33], Kenya 71%-91% [34], Gambia 36.7% [35], Senegal 78.4% [36] and in South Africa 58%-88% [37]. The

**Table 5. Factors of timely completion of vaccination among Ethiopian women using EDHS-2019 data.**

| Variables | Null model | Model I | Model II | Model III |
|---|---|---|---|---|
| **Current-breast feeding** | | | | |
| Yes | | 1.2 (0.85, 1.71) | | 1.18(0.83,1.67) |
| No | | Reference | | Reference |
| **Place of delivery** | | | | |
| Health facility | | | 1.55(1.34, 1.9)* | **1.62 (1.25, 2.13)*** |
| Home | | | Reference | Reference |
| **ANC utilization** | | | | |
| Yes | | 1.5 (1.23, 1.86)* | | **1.63 (1.3, 2.04)*** |
| No | | Reference | | Reference |
| **Women age** | | | | |
| > = 35 years | | 1.91(1.37, 2.65)* | | **1.89 (1.35, 2.63)*** |
| 25–34 years | | 1.98 (1.6, 2.89)* | | **1.72 (1.33,2.21)*** |
| 15–24 years | | Reference | | Reference |
| **Wealth index** | | | | |
| Richest | | 3.39 (2.35, 4.79)* | | **2.71 (1.86, 2.94)*** |
| Richer | | 1.7(1.16, 2.49)* | | 1.45 (0.98, 2.14) |
| Middle | | 1.4(0.96, 2.05) | | 1.31(0.89, 1.91) |
| Poorer | | 1.47 (1.03, 2.09) | | 1.37 (0.96, 1.95) |
| Poorest | | Reference | | Reference |
| **Sex of household the head** | | | | |
| Male | | 1.15 (0.88, 1.49) | | 1.12 (0.86, 1.45) |
| Female | | Reference | | Reference |
| **Educational status** | | | | |
| higher education | | 1.58 (1.06,2.39) | | 1.45 (0.96, 2.18) |
| Secondary education | | 1.48(1.04, 2.120 | | 1.35(0.94, 1.44) |
| Primary education | | 1.1(0.85, 1.42) | | 1.03(0.79,1.34) |
| No education | | Reference | | Reference |

possible reason for this discrepancy might be associated with the situation in Ethiopia during this period. Ethiopia has been in trouble since 2016. So the conflict in some parts of the country negatively affects health services, including vaccination practices. The other possible reason for being less than other African and Asian countries might be the socioeconomic difference between the countries. But it was higher than a study conducted in northwest Ethiopia in Menz Lalo district (6.2% [3]. This might be due to a study in northwest Ethiopia in the Menz Lalo district conducted only on Penta vaccine 1–3 and measles vaccine doses, but in the current study all vaccine doses are considered.

The spatial distribution of timely completion of vaccination in Ethiopia was non-randomly distributed across the regions, with a global Moran's Index test value of 1.114 (p-value <0.001). This means the timely completion of vaccinations in Ethiopia was statistically clustered in the regions. A statistically significant high proportion of timely completion areas were clustered in the eastern part of Amhara, the south part of Afar, Addis Ababa, and Oromia. Whereas the cold spot areas of timely completion of vaccination in Ethiopia were the southern people's nations and nationalities. The possible reason for this spatial variation of timely vaccination across the region of Ethiopia might be the variation in the knowledge of appropriate vaccination schedules, access to the health service, the variation of infrastructure to get the vaccines, and the variation of security problems across the enumeration areas. This may also be the variation counselling service about the appropriate schedule of vaccines across the region of Ethiopia.

Moreover, the spatial interpolation analysis (spatial ordinary kriging) predicted that highly practiced regions for timely completion of vaccination were Addis Ababa and Diredawa compared to the other parts of the regions. Again, in the Sat Scan analysis, the most likely clustered area of timely completion of vaccinations was in Diredawa, which was 3.68 times higher than outside the window. This might be because of the difference in the awareness level of timely vaccinations between urban and rural parts of Ethiopia [10].

The study also showed that the timely completion of vaccinations among women who had a history of ANC follow-up was higher than that of women without a history of ANC follow-up. This finding is supported by a study conducted in Tanzania [30], southeast Ethiopia of Sinana District [38], Central Ethiopia of Ambo District [39], and northern Shewa Oromia Ethiopia [3]. This might be because women who have ANC follow-up get more information, education, and counselling services about timely vaccination practices than those who have no contact with healthcare providers. Similarly, women who gave birth at health facilities were more likely to practice timely completion of vaccination than those who gave birth at home. This is supported by a study conducted in the northern Shewa zone of the Oromia regional state in Ethiopia. Demographic health data for Ethiopia (36), and the Kutaye district of Ethiopia [20]. The possible reason for this finding is that women who give birth at health facilities have more chances to get information and counselling about timely vaccination practices [20].

In the current study, the women's wealth index also affects timely vaccination practice, and thus the odds of timely completion of vaccination were higher among women who were richest than the poorest women. This is supported by a study conducted in Gahanna [10]. This is because the socioeconomic status of women is the most common challenge to vaccination access [40]. The other possible reason might be that better economic status positively affects the health-seeking behaviour of women, and they can access the service in a better way and easily [41]. Furthermore, the age of women affects the timely completion of vaccination practices. Women aged 25–34 were more likely to get timely vaccinations than women aged 15–24. This is due to older women being more likely to utilize the health service [42]. The other possible reason might be that women in this age group are more likely to have life experience with the schedule of vaccination and the importance of timely vaccination than young women. The

strength of the study was that the data was from a national-based survey, and a large sample size made for more powerful evidence. However, the study has its own limitations; it does not show a cause-and-effect relationship, and variables that could be important to the outcome were missed because it is secondary data. Additionally, the current study used the geographic coordinates of clusters (2 kilometers for urban areas and 5 kilometers for most clusters in rural areas), making it difficult to estimate the cluster effect in the spatial analysis.

## Conclusion

The prevalence of timely completion of vaccination was low in Ethiopia, and the spatial distribution of timely completion of vaccination in Ethiopia was non-randomly distributed across the regions. The factors associated with the timely completion of vaccinations were ANC follow-up, place of delivery, age of the participant, and wealth index. We recommend expanding facility delivery, antenatal care services, and empowering women to scale up timely vaccination in Ethiopia.

## Supporting information

**S1 Fig. PRISMA 2020 flow diagram for new systematic reviews which included searches of databases and registers only.**
(DOCX)

## Acknowledgments

The authors acknowledged the DHS international program for providing the data set.

## Author Contributions

**Conceptualization:** Muluken Chanie Agimas, Aysheshim Kassahun Belew, Mekonnen Sisay, Lemlem Daniel Baffa, Moges Gashaw, Esmael Ali Muhammad, Zeamanuel Anteneh Yigzaw, Berhanu Mengistu.

**Data curation:** Muluken Chanie Agimas, Aysheshim Kassahun Belew.

**Formal analysis:** Muluken Chanie Agimas, Mekonnen Sisay, Zeamanuel Anteneh Yigzaw.

**Investigation:** Muluken Chanie Agimas, Mekonnen Sisay.

**Methodology:** Mekonnen Sisay, Lemlem Daniel Baffa, Moges Gashaw, Berhanu Mengistu.

**Software:** Muluken Chanie Agimas, Aysheshim Kassahun Belew, Zufan Yiheyis Abriham, Esmael Ali Muhammad, Zeamanuel Anteneh Yigzaw, Berhanu Mengistu.

**Supervision:** Muluken Chanie Agimas, Berhanu Mengistu.

**Validation:** Muluken Chanie Agimas, Moges Gashaw, Esmael Ali Muhammad, Berhanu Mengistu.

**Visualization:** Moges Gashaw, Zufan Yiheyis Abriham, Zeamanuel Anteneh Yigzaw.

**Writing – original draft:** Muluken Chanie Agimas, Aysheshim Kassahun Belew, Zufan Yiheyis Abriham, Esmael Ali Muhammad.

**Writing – review & editing:** Muluken Chanie Agimas, Zufan Yiheyis Abriham.

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
