## [Decision Letter · Decision Letter 0]

12 Dec 2023

PONE-D-23-12639Spatial variations and determinants of timely completion of vaccination in Ethiopia using further analysis of EDHS 2019 data: Spatial and multilevel analysis.PLOS ONE

Dear Dr. muluken agimas chanie,

Thank you for submitting your manuscript to PLOS ONE. After careful consideration, we feel that it has merit but does not fully meet PLOS ONE’s publication criteria as it currently stands. Therefore, we invite you to submit a revised version of the manuscript that addresses the points raised during the review process.

We look forward to receiving your revised manuscript.

Kind regards,

Giuseppe Di Martino

Academic Editor

PLOS ONE

Journal Requirements:

4. Please include your tables as part of your main manuscript and remove the individual files. Please note that supplementary tables (should remain/ be uploaded) as separate "supporting information" files.

5. We note that Figures 3, 5, 6 & 8 in your submission contain [map/satellite] images which may be copyrighted. All PLOS content is published under the Creative Commons Attribution License (CC BY 4.0), which means that the manuscript, images, and Supporting Information files will be freely available online, and any third party is permitted to access, download, copy, distribute, and use these materials in any way, even commercially, with proper attribution. For these reasons, we cannot publish previously copyrighted maps or satellite images created using proprietary data, such as Google software (Google Maps, Street View, and Earth). For more information, see our copyright guidelines: http://journals.plos.org/plosone/s/licenses-and-copyright.

 a. You may seek permission from the original copyright holder of Figures 3, 5, 6 & 8  to publish the content specifically under the CC BY 4.0 license. 

Reviewers' comments:

Reviewer's Responses to Questions

**Comments to the Author**

1. Is the manuscript technically sound, and do the data support the conclusions?

Reviewer #1: Yes

Reviewer #2: Yes

2. Has the statistical analysis been performed appropriately and rigorously? 

Reviewer #1: Yes

Reviewer #2: No

3. Have the authors made all data underlying the findings in their manuscript fully available?

Reviewer #1: Yes

Reviewer #2: Yes

4. Is the manuscript presented in an intelligible fashion and written in standard English?

Reviewer #1: Yes

Reviewer #2: No

5. Review Comments to the Author

Reviewer #1: The data source and permission is well.stated in the article

Line 52 The ANC follow-up was not fully written

Line 53 The age cut off was not written

Figure 1 The legend " not timely completion of vaccination..." does not read well

Figure 2 Measles was misspelled

Reviewer #2: The authors studied “Spatial variations and determinants of timely completion of vaccination in Ethiopia using further analysis of EDHS 2019 data: Spatial and multilevel analysis”.

Comments to the Authors

Abstract

*The abstract section of your manuscript clearly justifies the study, however, the study could benefit from inclusion of the aim of the study with justification in the introduction section of the abstract instead of separately putting it as objective. It the authors have made it based on the journal guideline please leave this concern.

*Abstract methods; the authors described about the methodology of the data source instead of describing the authors’ own study. This section has incorporated too much information. The authors need to consider that they are in abstract and need to come with only main points of the methodology. For example, use of AIC for model selection is very specific and should be clearly addressed in the method section of body of manuscript with its importance not in the abstract.

*Under result of your abstract you did not have mentioned anything about results of spatial analysis, which I strongly recommend you to mention it.

In general, an abstract of a study should briefly address the burden of the problem, need for study, relevance, aim, method, results and conclusion with recommendation. The study could benefit from modification of the abstract.

Introduction

The introduction section of this manuscript best describes the background information regarding the problem its burden. Taking that into account, “According to the World Health Organization, timely vaccination is the practice of administering the vaccine within the first birthday of a child. (1).” Please remove the period before the reference. All vaccines are not initiated at the day of birth. On that ground, the authors need to come with more comprehensive definition and recommendation of the “timely completion of vaccination” instead of timely initiation.

The manuscript could benefit from modification the introduction with incorporation of its requirements.

Methods

“The source population was all women of Ethiopia who had a live birth aged 12-35 months earlier in the national survey and women who had live birth aged 12-35 months in the selected enumeration was the study population.” What is your base for cutoff points or for your study subjects to fall in the age range of 12-35 months? Perhaps, from the dataset, if that was the case, clearly mention it. Could you replace “live birth” with “baby or child"? Live birth would be more appropriate if you were dealing with subjects immediately after birth.

*Page 6, line 114, “Timely completion of vaccination (Yes, No)” needs to be described well about your outcome determination with a standard or fact. For example, how did you categorize or dichotomize “yes” or "no"?

* Line 118-119, “current breast feeding, total numbers of live birth, ANC follow-up, numbers of ANC visit and current pregnancy,” Please rewrite the sentence. It does not make sense by its current state. Were “ANC follow-up” and “numbers of ANC visit” different variables.

*Multilevel analysis was not well described in the methods section. The authors need to thoroughly describe multilevel analysis and its important parameters reported in the results of the study. For example, what multilevel is? Why did the authors use multilevel instead of ordinary logistic regression and how were parameters (ICC, PCV and AIC) determined? Should be clearly stated here. Refer to https://doi.org/10.1016/j.vaccine.2023.11.007

What is the case to use AIC instead of deviance or DIC for model selection?

*Ethical considerations

“Because the data were secondary or EDHS data ethical clearance was not applicable rather online permission was obtained from DHS international.” Your ethical consideration is crystal clear, but reader needs more information about source for your data like URL and you should describe Ethical consideration used by DHS. You can refer to (https://doi.org/10.1016/j.puhe.2023.04.010).

Result

*Would you please explain whether you used false discovery rate correction while you performed hot spot analysis? If yes, please mention that. If you haven't used that, then would you please justify why you didn't use false discover rate correction (which is very important). If it can't be justified, it will be great if the hot spot analysis could be done again. You may consult following article for help: article doi: 10.1371/journal.pone.0275951

*The DHS displaces the geographic locations of the clusters for maintaining confidentialities of the respondents. For this reason, when there is the possibility of using specific geographic location after an analysis, appropriate buffer is recommended to plot in the map (reference: Burgert CR, Prosnitz D. Linking DHS household and SPA facility surveys: Data considerations and geospatial methods. Rockville, Maryland, USA: ICF International; 2014. Report No.: DHS Spatial Analysis Reports No. 10.). I think using an appropriate buffer in hot spot analysis is very important. Is it possible to take an appropriate buffer around only the statistically significant hot and cold spots (cluster)? Otherwise, readers will not be able to understand the area of hot and cold spots.

Discussion

I found your discussion consistent with your results and rigorous. The results of the study were addressed with rigorous discussion. However, the study could benefit from limitation of spatial analysis in addition to the limitations addressed in the discussion.

General comments

For this study to be scientifically sound, I recommend the authors address all the comments raised in this review and reviews from other reviewers.

6. PLOS authors have the option to publish the peer review history of their article (what does this mean?). If published, this will include your full peer review and any attached files.

Reviewer #1: No

Reviewer #2: No

---

## [Decision Letter · Decision Letter 1]

18 Mar 2024

Spatial variations and determinants of timely completion of vaccination in Ethiopia using further analysis of EDHS 2019 data: Spatial and multilevel analysis.

PONE-D-23-12639R1

Dear Dr. Muluken Chanie Agimas,

We’re pleased to inform you that your manuscript has been judged scientifically suitable for publication and will be formally accepted for publication once it meets all outstanding technical requirements.

Kind regards,

Giuseppe Di Martino

Academic Editor

PLOS ONE

Additional Editor Comments (optional):

Reviewers' comments:

Reviewer's Responses to Questions

**Comments to the Author**

1. If the authors have adequately addressed your comments raised in a previous round of review and you feel that this manuscript is now acceptable for publication, you may indicate that here to bypass the “Comments to the Author” section, enter your conflict of interest statement in the “Confidential to Editor” section, and submit your "Accept" recommendation.

Reviewer #2: All comments have been addressed

2. Is the manuscript technically sound, and do the data support the conclusions?

Reviewer #2: Yes

3. Has the statistical analysis been performed appropriately and rigorously? 

Reviewer #2: Yes

4. Have the authors made all data underlying the findings in their manuscript fully available?

Reviewer #2: Yes

5. Is the manuscript presented in an intelligible fashion and written in standard English?

Reviewer #2: Yes

6. Review Comments to the Author

Reviewer #2: The authors of the manuscript addressed all the comments forwarded by my review, though it appears the document needs some language improvements.

7. PLOS authors have the option to publish the peer review history of their article (what does this mean?). If published, this will include your full peer review and any attached files.

Reviewer #2: No

---

## [Editor Report · Acceptance letter]

27 Mar 2024

PONE-D-23-12639R1 

PLOS ONE

Dear Dr. Agimas, 

I'm pleased to inform you that your manuscript has been deemed suitable for publication in PLOS ONE. Congratulations! Your manuscript is now being handed over to our production team.

Kind regards, 

on behalf of

Dr. Giuseppe Di Martino 

Academic Editor

PLOS ONE